# The Diversity, Composition, and Metabolic Pathways of Archaea in Pigs

**DOI:** 10.3390/ani11072139

**Published:** 2021-07-20

**Authors:** Feilong Deng, Yushan Li, Yunjuan Peng, Xiaoyuan Wei, Xiaofan Wang, Samantha Howe, Hua Yang, Yingping Xiao, Hua Li, Jiangchao Zhao, Ying Li

**Affiliations:** 1Guangdong Provincial Key Laboratory of Animal Molecular Design and Precise Breeding, College of Life Science and Engineering, Foshan University, Foshan 528225, China; fdeng@fosu.edu.cn (F.D.); okhuali@aliyun.com (H.L.); 2School of Life Science and Engineering, Foshan University, Foshan 528225, China; lyushan0@163.com (Y.L.); yunjuanpeng@gmail.com (Y.P.); 3Department of Animal Science, Division of Agriculture, University of Arkansas, Fayetteville, AR 72701, USA; xw010@uark.edu (X.W.); xxw033@email.uark.edu (X.W.); smhowe@uark.edu (S.H.); jzhao77@uark.edu (J.Z.); 4State Key Laboratory for Managing Biotic and Chemical Threats to the Quality and Safety of Agro-Products, Institute of Agro-Product Safety and Nutrition, Zhejiang Academy of Agricultural Sciences, Hangzhou 310021, China; yanghua@zaas.ac.cn (H.Y.); ypxiaozj@hotmail.com (Y.X.)

**Keywords:** swine, archaea, energy metabolism, CAZyme genes, ARGs

## Abstract

**Simple Summary:**

Archaea is identified as the key link in the interaction between gut microbiota and host metabo-lism. Studies on human and mice have reported archaea, especially methanogenic archaea, makes an important impact on the energy harvesting capacity of the host by improving fermentation. But, in pigs, the metabolic potential of archaea at different production stages are still largely unknown. Herein, we re-analyzed 276 metagenomic samples to explore the diversity, composi-tion, and potential functions of archaea in pigs. The results showed significant regional variations in archaeal composition. Furthermore, the Metacyc pathway related to hydrogen consumption (METHANOGENESIS-PWY) was only observed in archaeal reads, and archaea may be involved in carbohydrate metabolism and de novo synthesis of some kinds of essential amino acid. Overall, metagenomic re-analysis revealed that the composition and functional potential of archaea in the swine gut and suggested that archaea may make an important function in pigs.

**Abstract:**

Archaea are an essential class of gut microorganisms in humans and animals. Despite the substantial progress in gut microbiome research in the last decade, most studies have focused on bacteria, and little is known about archaea in mammals. In this study, we investigated the composition, diversity, and functional potential of gut archaeal communities in pigs by re-analyzing a published metagenomic dataset including a total of 276 fecal samples from three countries: China (*n* = 76), Denmark (*n* = 100), and France (*n* = 100). For alpha diversity (Shannon Index) of the archaeal communities, Chinese pigs were less diverse than Danish and French pigs (*p* < 0.001). Consistently, Chinese pigs also possessed different archaeal community structures from the other two groups based on the Bray–Curtis distance matrix. *Methanobrevibacter* was the most dominant archaeal genus in Chinese pigs (44.94%) and French pigs (15.41%), while *Candidatus methanomethylophilus* was the most predominant in Danish pigs (15.71%). At the species level, the relative abundance of *Candidatus methanomethylophilus alvus*, *Natrialbaceae archaeon* XQ INN 246, and *Methanobrevibacter gottschalkii* were greatest in Danish, French, and Chinese pigs with a relative abundance of 14.32, 11.67, and 16.28%, respectively. In terms of metabolic potential, the top three pathways in the archaeal communities included the MetaCyc pathway related to the biosynthesis of L-valine, L-isoleucine, and isobutanol. Interestingly, the pathway related to hydrogen consumption (METHANOGENESIS-PWY) was only observed in archaeal reads, while the pathways participating in hydrogen production (FERMENTATION-PWY and PWY4LZ-257) were only detected in bacterial reads. Archaeal communities also possessed CAZyme gene families, with the top five being AA3, GH43, GT2, AA6, and CE9. In terms of antibiotic resistance genes (ARGs), the class of multidrug resistance was the most abundant ARG, accounting for 87.41% of archaeal ARG hits. Our study reveals the diverse composition and metabolic functions of archaea in pigs, suggesting that archaea might play important roles in swine nutrition and metabolism.

## 1. Introduction

Before Dr. Carl Woese and his colleagues separated archaea (named “Archaebacteria” at that time) from the bacteria domain based on phylogenetic analysis of ribosomal RNA sequences, archaea were considered a subgroup of bacteria [1]. To highlight the differences between archaea and bacteria, Dr. Carl Woese formally proposed the name of “Archaea” in his publication [2]. With the advancement of research technologies and tools, a large number of novel archaea and their characteristics have been discovered.

Archaea have highly diverse energy sources and unique metabolic characteristics and cell physiology, making it possible for them to live in various extreme environments, such as extreme temperature [3], high-salt [4], extreme alkaline [5], or acidic environments [6]. At the same time, archaea were also detected in various non-extreme ecosystems, including soil, ocean, lake water, and habitats associated with human and animal hosts, and were confirmed to make significant contributions to the ecological cycle by previous studies [7,8]. As a “young” group with unique natural capabilities and biological characteristics, archaea have attracted increasing research interest in the past decade.

Unlike bacteria, archaea are one of the least studied and least understood members of the human microbiota. One of the reasons is that archaea only account for a very small proportion of the microbiota. A recent study among East Asians living in South Korea showed that archaea were detected in 42.47% (381/897) and constituted 10.24 ± 4.58% of total bacteria and archaea among positive subjects [9]. However, the relatively small proportion of archaea does not mean they are not important. For example, some species of archaea have been reported to reduce Trimethylamine-N-oxide (TMAO), a harmful product associated with several diseases [10], through methanogenesis. In addition, methanogenic archaea can remove H_2_, the end product of bacterial fermentation, to increase the efficiency of fermentation and improve the energy harvesting capacity of the host [11]. Two studies performed on adults and children report that hydrogen-utilizing methanogenic archaea are co-enriched with H_2_-producing bacteria, *Prevotella* species, in the gut of obese humans [12]. Furthermore, human gut colonization by archaea has been associated with several other gastrointestinal and metabolic diseases, such as chronic constipation [13], inflammatory bowel disease [14], and colorectal cancer [15].

In pigs, the major ecological niches harboring archaea along the gastrointestinal tract (GIT) include the cecum, colon, and rectum, according to Gresse et al. [16]. The genus *Methanobrevibacter* is predominant with a relative abundance of 57–100% of the archaeal communities in pigs and sows [16,17,18,19]. Additionally, the archaeal composition in the swine GIT is dynamic, according to Su et al. [17] and Federici et al. [20], and diet-driven [21]. These exploratory studies provided important information on archaea in pigs, however, their sample sizes were small, and the composition and metabolic potential of archaea at different production stages in pigs are still largely unknown.

Xiao et al. [22] established a swine gut microbiome reference gene catalog with metagenome deep sequencing of fecal samples from 287 pigs. In this study, we re-analyzed their data and investigated the diversity, composition, and potential functions of archaea in pigs.

## 2. Materials and Methods

### 2.1. Obtaining Data (Metadata (from Nature Microbiology))

A total of 287 fecal samples collected from China (*n* = 87), France (*n* = 100), and Denmark (*n* = 100) were sequenced using shotgun metagenomic sequencing technology by Xiao et al. [22], of them, 10 samples from China were omitted because of low sequencing depth, and we downloaded 276 of these sequences from the European Nucleotide Archive (ENA) under accession code PRJEB11755. We filtered 11 samples due to their low sequencing depths. In addition, we downloaded the predicted gene catalog generated by Liang et al., from GiGaDB (http://gigadb.org/dataset/view/id/100187/token/F4CDHYruxobOKmsE, accessed on 11 November 2020) to annotate the CAZy genes.

### 2.2. Raw Reads Pre-Processing

Downloaded raw reads were filtered using Kneaddata v0.7.2 (https://huttenhower.sph.harvard.edu/kneaddata/), an integrated pipeline used to perform quality control and remove host contamination from shotgun metagenomic data. In brief, reads were trimmed using Trimmomatic v0.39 [23] with parameter SLIDINGWINDOW:4:20 and MINLEN:60. Potential host contamination was identified and removed by mapping all reads to the swine reference genome (Scrofa 11.1, ftp://ftp.ncbi.nlm.nirefseq/vertebrate_mammalian/h.gov/genomes/Sus_scrofa/representative/GCF_000003025.6_Sscrofa11.1) using bmtagger v.3.102.4 [24] with default parameters. The number of clean metagenomic reads for fecal samples was rarefied to 16,230,988 paired reads for each sample for a consistent sequencing depth by random subsampling.

### 2.3. Taxonomic Profiling and Diversity

Kraken2 [25], a software used for short reads classification with the lowest common ancestor (LCA) algorithms, was used to classify clean short reads against the bacterial genome and archaeal genome databases. The bacterial genome database was downloaded using the Kraken2 standard command, which downloads NCBI taxonomic information and complete bacterial genomes in RefSeq (1 July 2020) [26]. Then, clean reads were classified as bacterial taxa using Kraken2, and reads identified as bacteria were retained for downstream analysis. The method used for archaeal classification was slightly different compared with that of bacteria. The archaeal genome database was built using Kraken2 from two sources: (1) RefSeq (1 July 2020) and (2) a curated archaeal reference dataset [27]. Similarly, Kraken2 was used to assign clean reads into proper archaeal taxa based on this archaeal database, and archaeal reads were separated and retained.

Reads count table of archaea and bacteria at the species level were entered into QIIME2 [28] platform to rarefy and calculate alpha diversity (Shannon Index) and beta diversity (Bray–Curtis).

### 2.4. CAZyme and ARG Analysis

HUMAnN3 [29] pipeline was used to characterize the metabolic pathways of archaea and bacteria using the MetaCyc database.

CAZyme family annotation was performed on the downloaded gene catalog generated by Xiao et al. via alignment with the dbCAN database [30] by HMMER3 software [31]. The genes identified as CAZy genes were extracted as target genes for further quantitative analysis. Salmon (quant) [32] was run on separated archaeal reads to quantify the archaeal CAZy genes in each sample, and a similar analysis was run on the separated bacterial reads.

Deeparg-SS [33] pipeline was adapted to predict antibiotic resistance genes in both bacterial and archaeal reads with the following critical parameters (--min-prob 0.8 --arg-alignment-identity 30 --arg-alignment-evalue 1e-10).

### 2.5. Statistical Analysis

For all analyses, statistical significance was determined at *p* < 0.05 and *p* < 0.001. Kruskal–Wallis (pairwise) was performed to explore differences in alpha diversities among different countries. Analysis of similarities (ANOSIM) was carried out to test the statistical significance of beta diversity among different countries. Kruskal–Wallis (pairwise) and ANOSIM were run in QIIME2.

## 3. Results

### 3.1. The Diversity of Archaea in Pigs

Significant differences in archaeal alpha diversity between different countries were observed. Pairwise comparisons of countries showed that archaeal diversity (Shannon Index) of Chinese pigs were significantly lower than that of the French (Figure 1A: Kruskal–Wallis test, *p* < 0.001) and Danish groups (*p* < 0.001). No differences in the swine archaeal alpha diversity were observed between French and Danish (*p* = 0.918) pigs. Beta diversity of the archaeal communities was visualized on a principal coordinate analysis (PCoA) plot based on the Bray–Curtis distance matrix. Chinese pigs had significantly different archaeal communities from Danish (ANOSIM, *p* < 0.001, R = 0.808) and French (ANOSIM, *p* < 0.001, R = 0.634) pigs as demonstrated on the PCoA plot (Figure 1B). However, the archaeal community structures of the Danish and French pigs could not be distinguished (Figure 1B; ANOSIM, *p* < 0.001, R = 0.109).

### 3.2. The Archaeal Composition in Pigs

At the genus level, *Methanobrevibacter*, with an overall average relative abundance of 23.02%, was the most abundant archaeal genus (Figure 2), followed by *Candidatus methanomethylophilus*, which had a relative abundance of 10.52%. Of note, although *Methanobrevibacter* is considered the most dominant archaeal genus of pigs in general, it was ranked the second most abundant in Danish pigs. Additionally, *Methanobrevibacter* was significantly more abundant in Chinese pigs with the relative abundance of 44.94% than in both Danish (13.97%) and French pigs (15.41%), while *Candidatus methanomethylophilus* was decreased in Chinese pigs (0.98%) compared to Danish (15.71%) and French (12.59%) pigs.

At the species level, the mean relative abundance of the top 15 species is shown in Figure 3. The gut microbiome of pigs is dominated by *Natrialbaceae archaeon* XQ INN 246 (mean relative abundance, 10.27%), followed by *Candidatus methanomethylophilus alvus* (9.57%), *Methanobrevibacter gottschalkii* (6.40%), and *Methanobrevibacter smithii* (3.42%). The most abundant species in different countries seem to be inconsistent, as well. The relative abundance of *Candidatus methanomethylophilus alvus*, *Natrialbaceae archaeon* XQ INN 246, and *Methanobrevibacter gottschalkii* were the highest in Danish, French, and Chinese pigs with an average relative abundance of 14.32, 11.67, and 16.28%, respectively.

### 3.3. The metabolic Pathways of the Swine Archaea

To investigate the possible roles of gut archaea in swine, MetaCyc metabolic pathway profiles based on archaeal reads were investigated using HUMAnN3. A total of 77 MetaCyc pathways were observed, with 12 of these MetaCyc pathways present in more than 60% of fecal samples (Appendix A). Figure 4A shows the top six most abundant MetaCyc pathways. The most abundant MetaCyc pathway was involved in pyruvate fermentation to isobutanol (PWY-7111). Pathways of VALSYN-PWY and ILEUSYN-PWY ranked as second and third and were related to the biosynthesis of two essential amino acids, L-valine and L-isoleucine, respectively. Interestingly, these pathways, PWY-7111, VALSYN-PWY, and ILEUSYN-PWY, were also among the top 10 most abundant MetaCyc pathways in bacteria (Figure 4B, Appendix A). Furthermore, the pathway related to hydrogen consumption (METHANOGENESIS-PWY) was only observed in archaea and not in bacteria (Appendix A). In contrast, the pathways involved in hydrogen production (FERMENTATION-PWY and PWY4LZ-257) were only observed in bacteria and not in archaea (Figure 5).

We next profiled the archaea-annotated CAZyme gene families in the swine gut. A total of 297 CAZyme gene families were identified from archaeal reads, out of which 99 were detected with a sample coverage of more than 60%. The three major CAZyme families were glycoside hydrolases (GH, 54/99), glycoside transferases (GT, 24/99), and carboxylesterase (CE, 11/99). The most abundant CAZyme category was AA3 (auxiliary activity family AA3), followed by GH43 and GT2 (Figure 6A). For comparison, we also investigated bacterial CAZyme profiles (Figure 6B) and found that six of the top 10 CAZyme families of archaea and bacteria were shared, including AA3, GT2, AA6, GT41, GT5, and GH13.

Finally, we examined the ARGs in the swine gut archaea. In total, 13 classes of ARGs were detected from archaeal reads in all samples. On average, 0.035% of ARG hits were obtained from archaeal reads. The ARG class of multidrug resistance represented the vast majority (87.41%) of ARG hits; of the ARG class of multidrug resistance, the proportion of rpoB2 gene ranged from 87.18 to 100%. The other ARG classes with more than 1% relative abundance included peptide (5.07%), tetracycline (2.29%), mupirocin (1.69%), and unclassified (1.15%). The ARG type of multidrug class was significant enriched in Chinese pigs (Kruskal-Wallis test, *p* < 0.01), and the relative abundance of the multidrug class in Danish pigs is significantly higher than that in French pigs (Kruskal–Wallis test, *p* < 0.01). The boxplots of the top six ARG genes are shown in Figure 6C. In contrast, 24 ARGs were identified in bacterial reads, and 0.144% of bacterial reads were identified as ARG hits. The most abundant ARG class in bacteria was tetracycline, accounting for 40.59% of the total ARG hits (Figure 6D), while the class of multidrug ranked second with 19.39%, followed by MLS (17.73%), beta-lactam (7.46%), peptide (5.33%), and aminoglycoside (4.63%).

## 4. Discussion

Although archaea are an essential class of gut microorganisms in humans and animals, they have been neglected for many years. A few studies have examined archaeal composition by 16S rRNA sequencing methods; however, most studies only focused on the composition of methanogenic archaea of gut archaea. In fact, there are many non-methanogenic archaea such as *Thermococcus*, *Halorubrum*, and *Halococcus* in the swine gut. Furthermore, only a small number of gut archaeal strains were isolated from pig feces [34]. Therefore, compared to bacteria, very limited information is available for the archaeal composition in pigs, and even less is known about the archaeal metabolic pathways in pigs. In this study, by re-analyzing the swine gut metagenome of a large number of pigs from three countries, we characterized the swine gut archaeal diversity and metabolic pathways, which improves our understanding of the potential roles of archaea in mammals. However, it is also worth noting that the lower abundant archaeal genes detected by metagenome analysis could be highly expressed when examined using metatranscriptomic approaches.

Regarding archaeal diversity, Chinese pigs had archaeal communities significantly different from Danish and French pigs in both alpha and beta diversity, consistent with the bacterial communities described by Xiao and colleagues [22]. Chinese pigs were fed two types of antibiotics, including fluorine benzene (400 g/ton) and penicillin (600g/ton), which may cause a significant reduction in microbial richness. However, most archaea were not sensitive to penicillin and fluorine benzene because of the absence of drug targets [35], which means that the administration of antibiotics could not fully explain the decrease of archaeal richness in Chinese pigs. However, it is well known that antibiotics affect bacterial community diversities, which might in turn have influenced the archaeal community due to bacterial–archaeal interactions. In addition, geography, environment, diet, sex, and breed are confounding factors that might contribute to the difference in alpha and beta diversity of the swine gut archaeal diversity between the three countries for archaeal composition, *Methanobrevibacter* and *Candidatus methanomethylophilus* were the top two most abundant genera. *Methanobrevibacter* is a hydrogen-oxidizing genus that produces methane from carbon dioxide and hydrogen [36], while *Candidatus methanomethylophilus* synthesizes methane through the reduction of trimethylamine and methanol [37]. *Methanobrevibacter* was the most abundant archaeal genus in general, which was in line with previous studies [16]. It should be noted that variations in the abundance of *Methanobrevibacter* between countries were observed. It was ranked the second most abundant genus in the Danish group with a relative abundance of 13.97%, following *Candidatus methanomethylophilus* (relative abundance of 15.71%). The archaeal composition based on metagenomics in this study was different from that revealed by 16S rRNA or mcrA genes in other studies. Mi and colleagues found that the genus *Methanobrevibacter* was the dominant archaeal taxon with an average relative abundance of 57% in the large intestine of finishing pigs [19]. Furthermore, *Methanobrevibacter* was also reported as the overwhelmingly dominant archaeal genus and accounted for 95.01–100% of all archaeal reads in pigs in other studies [17,18]. Many factors, such as environment, diet, breed, and age, might contribute to the discrepancies in the archaeal composition between these studies. It is worth noting that technology (i.e., shotgun metagenomics sequencing vs. 16S rRNA sequencing) is likely another important confounder. Amplification bias is an inevitable disadvantage to accurately quantify bacterial and archaeal cells [38]. Of particular note, compared with bacteria, the domain archaea remains poorly understood, and there is a lack of sufficient reference sequences to design primers properly, which causes many archaeal taxa to remain undetected [39]. In addition, the lack of archaeal reference sequences not only affects proper primer design, but also hinders the annotation of archaeal genes, resulting in the elusive view of archaeal functions in pigs.

Previous studies on animals and humans have revealed that archaea were involved in energy metabolism through the consumption of hydrogen and carbon dioxide to produce methane [11,40]. Consistent with those studies, the metabolic pathway related to methanogenesis from H_2_ and CO_2_ (METHANOGENESIS-PWY) was only found in archaeal reads. In contrast, hydrogen production pathways were only found in bacterial reads and not in archaeal reads. These results suggest that there is a complementary relationship between archaea and bacteria, and archaea participated in energy metabolism in pigs as well. Samuel and colleagues showed that methanogenic archaea were able to enhance caloric harvest and weight gain by methods other than hydrogen utilization in humans, but the detailed mechanism is unclear [40]. We inferred that archaea have the potential to improve the growth performance of pigs through the synthesis of essential amino acids. In our study, L-valine and L-isoleucine-related pathways were ranked as the top MetaCyc pathways of archaea. L-valine and L-isoleucine are essential amino acids and could improve the growth performance of pigs at multiple stages of growth [41,42]. Of note, the two pathways were also abundant in bacteria. Further studies are desired to elucidate the proportion and magnitude of archaeal contribution to amino acid biosynthesis in pigs.

Interestingly, a large number of CAZyme gene families were also detected in archaeal reads. CAZyme genes are involved in carbohydrate metabolism and are important in the absorption and utilization efficiency of energy for the host. CAZyme gene families abundance in the host gut were confirmed association with growth performance in several domestic animals [43,44,45]. In general, bacteria in the mammal gut, especially species belonging to Firmicutes and Bacteroidetes, were considered the main providers of CAZyme genes for encoding enzymes to digest complex carbohydrates [46]. Nevertheless, recent publications revealed that archaea participate in carbohydrate metabolism through the production of various enzymes [47,48]. In our study, 297 CAZyme gene families were detected in archaea reads, and six of the top 10 CAZyme families of archaea and bacteria were shared. These CAZyme gene families of archaea may be useful supplements for carbohydrate digestion in swine. Therefore, our study suggests that, in addition to H_2_ depletion, archaea might be directly involved in energy metabolism in the pig gut.

Bacterial ARGs have been investigated in many studies in different animals. However, little is known about ARGs in archaea. In this study, we also explored the archaeal ARGs in pigs in addition to metabolic pathways. The ARG classes of tetracycline, multidrug, and MLS were relatively abundant in bacterial reads of the 287 pigs. However, the ARG class of multidrug resistance represented the overwhelming majority at 87.41% of ARG hits in archaeal reads of these pigs. RpoB2 gene accounted for 87.18–100% of the multidrug resistance type. The rpoB2 gene confers resistance to rifamycin through influencing Rif binding, which decreases antibiotic activity of rifamycin. The effects of low-dose antibiotics on archaeal ARGs in pigs, the generation and spread of ARGs among archaeal and between domains need further investigation.

In this study, we profiled archaeal composition and functional potential in pigs. Although amplification bias was avoided by using shotgun metagenomics data, there are two limitations of our study. First, our study only included a limited number of archaeal reference genomes. Therefore, numerous archaea with no reference genomes could not be identified, which can underestimate the richness of archaea and its functional diversity. Second, compared with bacteria, the relative abundance of archaea is lower by about 1%. Therefore, higher sequencing depth was demanded to quantify archaea accurately.

Although we demonstrated archaea had functional potentials, further studies are needed to link archaeal species or strains with the growth performance of pigs. In addition, methane, the production of methanogenic archaea, is a greenhouse gas with environmental concerns. Thus, the modulation of archaeal communities to maintain a balance between animal production (e.g., growth-promoting functions) and environmental protection (e.g., methane emission) is critical.

## 5. Conclusions

In conclusion, our study revealed the composition and functional potential of archaea in the swine gut based on the re-analysis of metagenomic data. We have demonstrated different predominant archaeal genera and species in different swine populations. In terms of functional potential, METHANOGENESIS-PWY related to methanogenesis from H_2_ and CO_2_ was only found in archaeal reads, which indicated that archaea were involved in energy metabolism by depleting H_2_ in the swine gut. Moreover, archaea shared several amino acid synthesis pathways and the main CAZyme gene families with bacteria, which suggested archaea are potentially involved in amino acid synthesis and energy metabolism.

## Figures and Tables

**Figure 1 animals-11-02139-f001:**
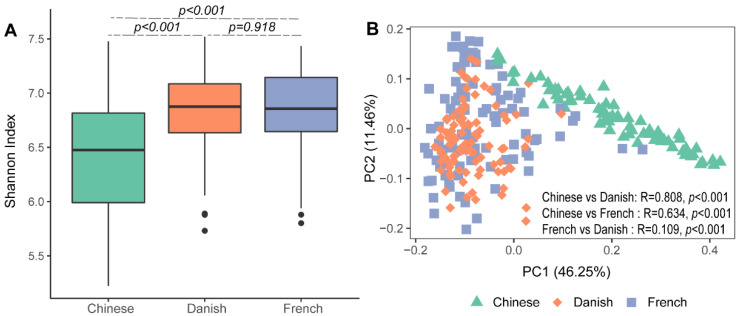
Swine archaeal alpha and beta diversities between three countries. Alpha diversity of archaeal communities in pigs was measured using the Shannon Index (**A**). Beta diversity was illustrated on a principal coordinate analysis (PCoA) plot based on Bray–Curtis distance (**B**).

**Figure 2 animals-11-02139-f002:**
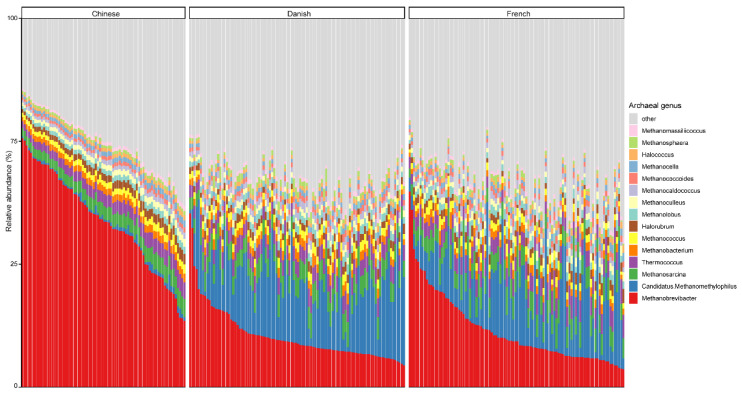
Relative abundance of the top 15 archaeal genera in swine fecal samples from different countries. Each column represents one pig, and each color represents one archaeal genus. The *y*-axis represents relative abundance ranging from 0 to 100%.

**Figure 3 animals-11-02139-f003:**
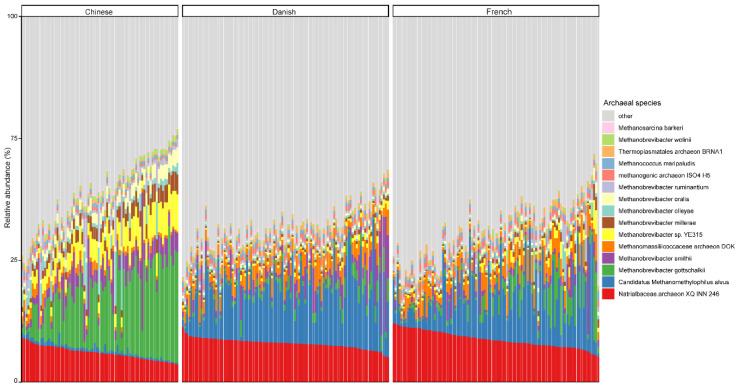
Relative abundance of the top 15 archaeal species in swine fecal samples from different countries. Each column represents one pig, and each color represents one archaeal species. The *y*-axis represents relative abundance ranging from 0 to 100%.

**Figure 4 animals-11-02139-f004:**
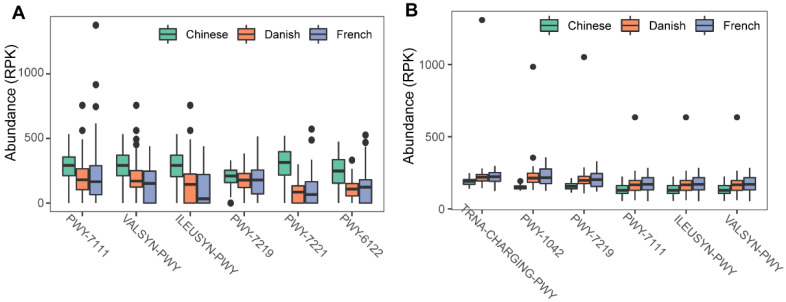
Summary of the predominant MetaCyc pathways of archaea and bacteria in pigs. The top six most abundant MetaCyc pathways of archaea (**A**) and bacteria (**B**) are displayed. In (**A**,**B**), the *y*-axis represents the number of reads mapped to reference genes involved in pathways. Data have been normalized to RPKM (reads per kilobase of transcript per million mapped reads).

**Figure 5 animals-11-02139-f005:**
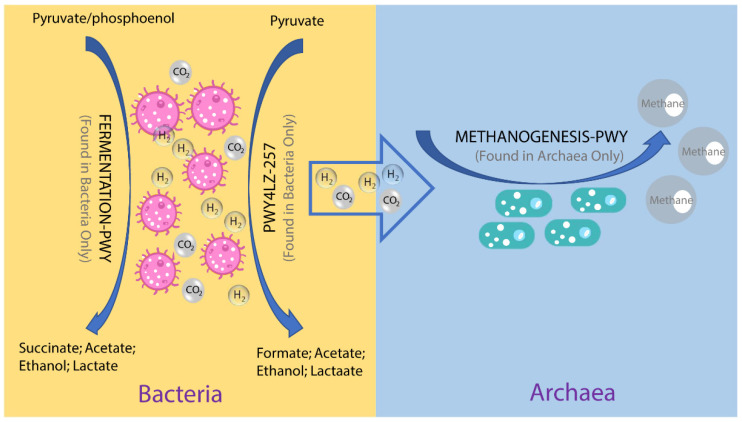
Overview of the hydrogen production pathways of bacteria and consumption pathway of archaea.

**Figure 6 animals-11-02139-f006:**
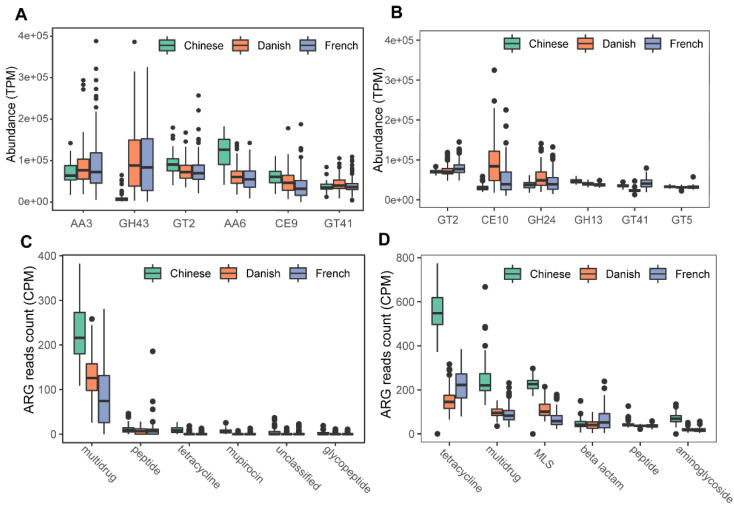
Summary of predominant CAZyme gene families (**A**,**B**) and antibiotic resistance genes (**C**,**D**) of archaea and bacteria in pigs. The top six CAZy gene families of archaea (**A**) and bacteria (**B**) were normalized by TPM (trans per million) using salmon software. ARG raw counts were identified using deeparg software. The ARG reads count of archaea (**C**) and bacteria (**D**) were normalized by CPM (counts per million).

## Data Availability

The data generated and analyzed during this study are included in this article.

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
