# Peer review of "The Diversity, Composition, and Metabolic Pathways of Archaea in Pigs"

_animals, 2021, doi:10.3390/ani11072139_

Round 1

Reviewer 1 Report

Animals Manuscript Draft

Manuscript Number: animals-1272378

Title: The diversity, composition, and metabolic pathways of archaea in pigs

Article Type: Research Article

Authors: Feilong Deng, Yushan Li, Yunjuan Peng, Xiaoyuan Wei, Xiaofan Wang, Samantha Howe, Hua Yang, Yingping Xiao, Hua Li, Jiangchao Zhao and Ying Li

The authors investigated the composition, diversity, and functional potential of gut archaeal communities in pigs by re-analyzing a published metagenomic dataset of 276 fecal samples from China (n=76), Denmark (n=100), and France (n=100). For alpha diversity (Shannon Index) of the archaeal communities, Chinese pigs were less diverse than Danish and French pigs (P<0.001) and Chinese pigs further possessed different archaeal community structures from the other two country groups based on the Bray-Curtis distance matrix. In terms of metabolic potential, the top three pathways in the archaeal communities included the MetaCyc pathway related to the biosynthesis of L-valine, L-isoleucine, and isobutanol. The pathway related to hydrogen consumption was only observed in archaeal reads, while the pathways participating in hydrogen production were only detected in bacterial reads. Archaeal communities also possessed carbohydrate-active enzyme (CAZyme) gene families, with the top five being: AA3, GH43, GT2, AA6, and CE9. In terms of antibiotic resistance genes (ARGs), the class of multidrug resistance was the most abundant ARG, accounting for 87.41% of archaeal ARG hits. The study reveals the diverse composition and metabolic functions of archaea in pigs, suggesting that archaea might play important roles in swine nutrition and metabolism.

General comments

The manuscript is generally a quite well written based on a profound re-analysis of an existing dataset.

The presentation of data in figures and tables (supplementary) is also clear and concise.

The reported data certainly highlight the importance of addressing the monogastric archaeal community and, not least, the metabolically capacity (in addition to methanogenesis) of archaea – in this case the archaea of monogastric pigs – instead of merely focusing on e.g. the bacterial microbiota members.

Thus, important with the indication of archaea being involved in e.g. synthesis of essential amino acids as well as energy metabolism (carbohydrate turnover) in addition to the major function as hydrogen scavengers. Still however, important to keep in mind that archaea makes up only a minor part (regarding cell numbers and total biomass) of the gut microbiota, which the authors also imply by using the expression ‘useful supplements’ of the following sentence (line 300-301): “These CAZyme gene families of archaea may be useful supplements for carbohydrate digestion in swine.”

The authors could, moreover, briefly address/discuss the use of a metagenomics approach compared to a metatranscriptomics approach; the genes /genetic capacity may be present, but to what degree are they/is it expressed.

The present manuscript is by nature quite descriptive, which is OK as such, since the presented data opens up for digging further into the role of archaea in monogastric animals (as well as in ruminants). However, I would maybe recommend the authors to reflect on and discuss some of the elements a bit further. For the topic of archaeal carbohydrate metabolism, could the authors maybe address some differences between the mainly lithoautotrophic gut methanogens and other archaea (addresses in some of the included references). Likewise, with respect to the antibiotic resistance genes, could it be somewhat further outlined what the multidrug resistance mechanisms found in archaea are? Unspecific efflux pumps or?

In conclusion, the findings of present manuscript are definitely of high importance and of interest for the scientific community working in the area of livestock (in particular monogastric) gut microbiology, animal health and performance as well as enteric methane mitigation, however, I urge the authors to take the general and specific comments into consideration.

Specific comments

  1. 177 and 181. The expression ‘one archaeal community’, could maybe be replaced by ‘one pig’
  2. 222. Here and elsewhere: can ‘multidrug resistance’ maybe be specified somewhat – can it be said mostly to signify e.g. efflux pumps, or…?
  3. 237-238. I would like the authors to address somewhat further what non-methanogenic gut archaea could be…?
  4. 251-252. Could this also have been an indirect effect of the antibiotics influencing the bacterial community, which may then have influenced the archaeal community due to bacterial-archaeal interactions
  5. 259-260. Maybe rephrase somewhat; sounds as if you directly compared amplicon (16S rRNA and mcrA genes) and shotgun (metagenomics) sequencing for the involved samples, which you did not.
  6. 269-271. Maybe worth mentioning, the lack of archaeal reference sequences not only affect potential primer design, but also annotation of archaeal genes.
  7. 275-276. Also just a matter of phrasing, but it would have been revolutionary, if a bacterial pathway for methanogenesis had been found, so I reckon this observation would (so far) go for any study.

Author Response

General comments

The manuscript is generally a quite well written based on a profound re-analysis of an existing dataset.

The presentation of data in figures and tables (supplementary) is also clear and concise.

The reported data certainly highlight the importance of addressing the monogastric archaeal community and, not least, the metabolically capacity (in addition to methanogenesis) of archaea – in this case the archaea of monogastric pigs – instead of merely focusing on e.g. the bacterial microbiota members.

Thus, important with the indication of archaea being involved in e.g. synthesis of essential amino acids as well as energy metabolism (carbohydrate turnover) in addition to the major function as hydrogen scavengers. Still however, important to keep in mind that archaea makes up only a minor part (regarding cell numbers and total biomass) of the gut microbiota, which the authors also imply by using the expression ‘useful supplements’ of the following sentence (line 300-301): “These CAZyme gene families of archaea may be useful supplements for carbohydrate digestion in swine.”

Response: We greatly appreciate the positive and encouraging comments.

The authors could, moreover, briefly address/discuss the use of a metagenomics approach compared to a metatranscriptomics approach; the genes /genetic capacity may be present, but to what degree are they/is it expressed.

Response: This is a great point. We totally agree that the presence and expression of certain genes could be very different when examined by using metagenome vs metatranscriptome. As suggested, we added a sentence in the discussion to address this comment “However, it is also worth noting that the lower abundant archaeal genes detected by metagenome analysis could be highly expressed when examined using metatranscriptomic approaches.”  (Page 8, line 250-252) In another manuscript that we are preparing we found that several archaeal species were highly expressed with a much higher abundance in the transcripts compared to gene counts. 

The present manuscript is by nature quite descriptive, which is OK as such, since the presented data opens up for digging further into the role of archaea in monogastric animals (as well as in ruminants). However, I would maybe recommend the authors to reflect on and discuss some of the elements a bit further. For the topic of archaeal carbohydrate metabolism, could the authors maybe address some differences between the mainly lithoautotrophic gut methanogens and other archaea (addresses in some of the included references).

Response: This is definitely a great point. However, it is very challenging to address this question. First, it takes a lot of time and computation power to re-analyze the data and to link the genes involved in carbohydrate metabolism to specific archaea. Second, there is a paucity of information related to archaeal carbohydrate metabolism, especially in monogastric animals. It might be a better time for such comparisons when more data are available, hopefully in the near future after we finish our global metagenome analysis of the swine gut microbiota. We will definitely make such comparisons in that manuscript.

 Likewise, with respect to the antibiotic resistance genes, could it be somewhat further outlined what the multidrug resistance mechanisms found in archaea are? Unspecific efflux pumps or?

Response: Great points!  We added the following to the text (page 9, lines 329-331), “RpoB2 gene accounted for 87.18% - 100% of the multidrug resistance type. The rpoB2 gene confers resistance to rifamycin through influence Rif binding, which decreases antibiotic activity of rifamycin

In conclusion, the findings of present manuscript are definitely of high importance and of interest for the scientific community working in the area of livestock (in particular monogastric) gut microbiology, animal health and performance as well as enteric methane mitigation, however, I urge the authors to take the general and specific comments into consideration.

Response: Thank you for these great comments!

Specific comments

177 and 181. The expression ‘one archaeal community’, could maybe be replaced by ‘one pig’

Response: We agree and have revised as suggested. Page 5, lines 179 and 183.

  1. Here and elsewhere: can ‘multidrug resistance’ maybe be specified somewhat – can it be said mostly to signify e.g. efflux pumps, or…?

Response: great point. We added the following to the text (page 9, lines 329-331), “RpoB2 gene accounted for 87.18% - 100% of the multidrug resistance type. The rpoB2 gene confers resistance to rifamycin through influence Rif binding, which decreases antibiotic activity of rifamycin”

237-238. I would like the authors to address somewhat further what non-methanogenic gut archaea could be…?

Response: We added the following to the text, page 8, lines 242-243 “In fact, there are many non-methanogenic archaea such as Thermococcus, Halorubrum and Halococcus in the swine gut.”

251-252. Could this also have been an indirect effect of the antibiotics influencing the bacterial community, which may then have influenced the archaeal community due to bacterial-archaeal interactions

Response : Great point. We agree and has added the following text to indicate this point in the Discussion section. “However, it is well known that antibiotics affect bacterial community diversities, which might in turn have influenced the archaeal community due to bacterial-archaeal interactions.”  Page 8, line 260-262.

259-260. Maybe rephrase somewhat; sounds as if you directly compared amplicon (16S rRNA and mcrA genes) and shotgun (metagenomics) sequencing for the involved samples, which you did not.

Response: Thanks for the advice. We rephrased this sentence “The archaeal composition based on metagenomics in this study was different from that revealed by 16S rRNA or mcrA genes in other studies” Page 8, line 273-275.

269-271. Maybe worth mentioning, the lack of archaeal reference sequences not only affect potential primer design, but also annotation of archaeal genes.

Response: Great point. We added the following in the text. “In addition, the lack of archaeal reference sequences not only affects proper primer design, but also hinders the annotation of archaeal genes, resulting in the elusive view of archaeal functions in pigs. ” Page 8, line 288-290.

275-276. Also just a matter of phrasing, but it would have been revolutionary, if a bacterial pathway for methanogenesis had been found, so I reckon this observation would (so far) go for any study.

Response: We agree and have rephrased the sentence as “Consistent with those studies, the metabolic pathway related to methanogenesis from H2 and CO2 (METHANOGENESIS-PWY) was only found in archaeal reads. ”  Page 9, line 293-295.

Reviewer 2 Report

“THE DIVERSITY, COMPOSITION, AND METABOLIC PATHWAYS OF ARCHAEA  IN PIGS”

The information and results presented in this manuscript bring a relevant contribution to the knowledge of archaea's composition and functional potential in the swine gut, which are still largely unknown. The work is professionally written and structured, with figures and explanations that help interpret the results obtained.

Minor changes:

Line 27. Set “Candidatus Methanomethylophilus” name in italics.

Line 88 to 91. Add how many samples were re-analyzed in each country evaluated.

Lines 141 and 142. I suggest this sentence can be removed, as it has already been specified that the 276 samples were re-analyzed in the material and methods.

Line 210. Add the CAZyme AA3 family name as it has not been mentioned previously.

Lines 275 to 272. Although the genera name indicates their roles, I suggest adding phrases about the functions of these two genera highlighted in the discussion might be interesting.

Author Response

The information and results presented in this manuscript bring a relevant contribution to the knowledge of archaea's composition and functional potential in the swine gut, which are still largely unknown. The work is professionally written and structured, with figures and explanations that help interpret the results obtained.

Response: We greatly appreciate the positive comments from this reviewer.

Minor changes:

Line 27. Set “Candidatus Methanomethylophilus” name in italics.

Response: Nice catch. Revised as suggested. Page 1, line 27.

Line 88 to 91. Add how many samples were re-analyzed in each country evaluated.

Response: Thanks for the suggestion. We now added the sample sizes in the text. “A total of 287 fecal samples collected from China (N=87), France (N=100), and Denmark (N=100) were sequenced using shotgun metagenomic sequencing technology by Xiao et al.[22], of them, 10 samples from China were omitted because of low sequencing depth, and we downloaded 276 of these sequences from the European Nucleotide Archive (ENA) under accession code PRJEB11755.” Page 2, line 88-92.

Lines 141 and 142. I suggest this sentence can be removed, as it has already been specified that the 276 samples were re-analyzed in the material and methods.

Response: we agree and have removed this sentence. Page 4, line 142-143.

Line 210. Add the CAZyme AA3 family name as it has not been mentioned previously.

Response: Thanks for your suggestion. We added AA3 family name of “(Auxiliary Activity Family AA3)”. Page 7, line 212-213.

Lines 255 to 272. Although the genera name indicates their roles, I suggest adding phrases about the functions of these two genera highlighted in the discussion might be interesting.

Response: Thanks for the suggestion. We have added a sentence to discuss their functions. Page 8, lines 265-269. “As for archaeal composition, Methanobrevibacter and Candidatus Methanomethylophilus were the top 2 most abundant genera. Methanobrevibacter is a hydrogen-oxidizing genus that produces methane from carbon dioxide and hydrogen[36], while Candidatus Methanomethylophilus synthesizes methane through the reduction of trimethylamine and methanol[37].”